# DO LLMS LEARN GRAPH REPRESENTATIONS WITHOUT CONTEXT?

## ABSTRACT

Large Language Models (LLMs) are trained on next-word prediction yet often appear to acquire structured knowledge beyond surface statistics. A central question is whether such internal representations emerge during zero shot learning without additional cues or only when explicit context is provided. We address this by training GPT-style models on paths sampled from synthetic and real-world graphs under two regimes: in-context learning, where subgraph information is provided, and zero shot learning, where only query nodes are given. We evaluate models through adjacency matrix reconstruction and linear probing of hidden activations. We find evidence that in-context learning models consistently recover graph structure and encode neighborhood information, while zero shot learning models fail to develop comparable representations.

## 1 INTRODUCTION

Large language models are neural networks that are based on the transformer architecture, these models are trained on the input sentences on a simple "next-word-prediction" task [1]Vaswani et al. (2023a); Brown et al. (2020c); Devlin et al. (2019). The models take user queries as input and generate predictions in response.

Two major techniques are used when generating predictions from these models: (1) In-context learning, where few-shot examples are provided alongside the query to guide the model toward better predictions, and (2) Zero shot learning, where only the input query is provided to the modelBrown et al. (2020b); Yang et al. (2024); Weber et al. (2023); Min et al. (2022); Han et al. (2023); Gozeten et al. (2025).

Despite being trained on such a simple task, these models demonstrate remarkable capabilities in understanding context from natural language text data. These are models are used for a plethora of tasks, including solving logic puzzles, writing and debugging computer programs, and answering general user queries Abdou et al. (2021); Li et al. (2021).

Generally in-context learning approaches have proven to provide better predictive capabilities than the zero shot learning approaches Wei et al. (2023); Brown et al. (2020a); Agarwal et al. (2024). However, how these capabilities emerge in these models while being trained on a simple next word prediction task, remains unclear. There are two main theories that attempt to explain working of these models : 1 ) These models only understand correlation of words in the sentences used in the training data, thus only learning the surface level statistics Bender et al. (2021). 2) These models do more than learn the surface level statistics and develop an internal representations for very simple concepts, such as color, direction, game state etc Li et al. (2024); Karvonen (2024); Vafa et al. (2024).

The world representation theory has primarily been explored through in-context learning models, where context such as partial game states is provided alongside the input Li et al. (2024); Karvonen (2024). In this scenario, it is difficult to determine whether learned representations emerge from the provided context or from the underlying training data.

---

[1]Modern LLMs may also be trained with reinforcement learning techniques, but this aspect is beyond the scope of this study. LLMs trained solely on next-word prediction also exhibit capabilities such as solving logic puzzles, reasoning, Wei et al. (2023); Brown et al. (2020a); Agarwal et al. (2024)

In this paper we investigate whether large language models develop internal representations of simple graph structures during zero shot learning. We study this problem by training models using the two approaches described above on the sequences generated from a graph. Then employ various probing techniques to determine whether the models have constructed internal graph representations from the training data.

To assess whether models have learned internal representations, we use the model's predictive capabilities to generate the adjacency matrix of the underlying graph, which we then compare it to the original adjacency matrix of the graph to calculate the reconstruction errors by the model. We also use linear probes to understand whether the activations of the trained model contained representation of the underlying graph.

Reconstruction errors and probing accuracy provide complementary perspectives on understanding the representation learned by the model. Reconstruction evaluates whether the model's outputs can recover the adjacency matrix, while probing tests whether hidden activations encode edge information independent of the output.

We observe that models trained using in-context learning produce fewer errors and are able to recover the internal structure of the graph, compared to the model trained using the zero shot learning approach. We find no evidence that the zero shot learning models learn internal representation of the underlying graph structure from training data. These findings are interesting and timely since they demonstrate that these models are only able to reconstruct the underlying structure when it is explicitly provided in the input.

## 2 RELATED WORK

**Large Language Models :** Large Language Models (LLms) are non-linear machine learning models that are built on the transformer architecture Vaswani et al. (2023b). These are trained on a huge corpora of dataset and have demonstrated remarkable capabilities to perform various tasks such as question answering, summarization, puzzle solving etc Devlin et al. (2019); Brown et al. (2020c). Their ability to generalize from patterns in data has made them a focal point of research in natural language processing and machine learning.

**World-Representation in LLMs:** Despite their success, LLMs are largely blackboxes and understanding their internal mechanisms remains a key challenge. One prominent research direction is the study of world-representations-the extent to which models learn internal representations of structured environments from the training data. In such studies, LLMs are trained on sequences generated from controlled environments, such as game boards, and researchers analyze the learned representations. Notable examples include investigations into games like Othello, Chess, and simplified spatial reasoning tasks Karvonen (2024); Li et al. (2024); Vafa et al. (2024). These studies explore whether LLMs encode underlying rules, states, or other abstract features of the environment.

**In-Context Learning and Zero Shot Learning :** Predictions from LLMs are typically generated using either in-context learning or zero shot learning approaches.

1) `In context learning :` In this setting, the model receives both the query and a few input examples. These examples guide the model toward more accurate predictions and reduce errors Brown et al. (2020b); Yang et al. (2024); Weber et al. (2023); Min et al. (2022); Han et al. (2023). In-context learning is often used to study internal representations: for instance, providing partial game states along with next moves allows researchers to analyze attention patterns and reconstruct the game-state representations learned by the model Li et al. (2024); Karvonen (2024).

2) `Zero Shot Learning:` Here, the model is trained on input sequences, but during prediction, no additional examples are provided. While simpler to deploy, this approach generally yields lower predictive accuracy than in-context learning and provides fewer cues for extracting internal representations Gozeten et al. (2025).

**Probing :** Probing is a standard methodology to investigate whether models encode specific feature or concepts in its activations. A probe is typically a classifier or regressor that takes the activations of a trained model as input and predicts a feature of interest, such as part-of-speech tags, syntactic structure, or game state Alain and Bengio (2016); Belinkov (2021); Krause et al. (2020). Probing has been widely used to explore both linguistic knowledge in LLMs and abstract representations in

structured environments, providing insight into what information is encoded and where it resides within the network.

## 3 PRELIMINARIES

### 3.1 GENERATING TRAINING AND VALIDATION DATA FROM THE GRAPH

A non-weighted graph is defined as $\mathcal{G} = (V, E)$ where $V$ are the vertices of the graph and $E$ represents the edges between the vertices. To generate training data, we generate random paths from the graph. Each sequence starts with the $S$ (the start node of the path), $D$ end node of the path and $L$ length of the path, followed by the corresponding path. The validation dataset is generated by sampling sub-paths from the sequences while ensuring that no $(S, D)$ pair appears in both training and validation datasets.

To encourage the model to learn when paths do not exist, we also include $(S, D)$ pairs with no valid path of length $L$ in both training and validation datasets. In such cases, instead of providing a path, we insert a special [NP] token, which signals the absence of a valid connection between the nodes. For in-context learning, the model input additionally includes a subgraph relevant to the current path. Full details on query generation are provided in the Appendix.

In this setup, the tuple $(S, D, L)$ serves as the query, and the corresponding path or the special [NP] token when no such path exists serves as a response to the query.

### 3.2 COMPUTING ERRORS DURING RECONSTRUCTION

Our models are trained to predict paths between node pairs in the graph. The predictions can be used to reconstruct an adjacency matrix, which serves as a proxy for the internal representation of the graph learned by the model.

For a given path length $L$, we sample $N$ $(S, D)$ pairs not present in the training dataset[2] and generate predictions for each pair. These predictions are used to construct the adjacency matrix. The reconstructed matrix is then compared with the original adjacency matrix to quantify errors:

$gt_e$: edges present in the original graph but absent in the reconstruction (indicating gaps or biases).

$pr_e$: edges present in the reconstruction but absent in the original graph (indicating hallucinations).

Ideally, both $gt_e$ and $pr_e$ should be close to zero, indicating accurate reconstruction without bias or hallucination.

A direct comparison against the full original adjacency matrix may overestimate errors, because sampled paths do not necessarily cover every edge of the graph. To address this, we construct a reduced "reference" adjacency matrix that contains only the edges present in the training data. The error metrics are then computed relative to this reduced matrix. This adjustment ensures that the evaluation measures how well the model captures the graph structure that was actually presented during training, rather than penalizing it for missing edges it had no opportunity to learn.

### 3.3 PROBING INTERNAL STATE

Probing allows us to determine whether specific structural information is encoded in the intermediate layers of a trained model, independently of its output predictions.

We focus on the existence of an edge between two nodes. Given a pair of nodes $(S, D)$, the probe is trained to predict whether an edge exists between S and E, i.e., whether $(S, D) \in \{edges\}$. The ability of a probe to recover this information from hidden activations indicates that the model has encoded neighborhood structure.

A linear probe can only succeed if the relevant information is linearly separable in the model's activation space. This ensures that high probe accuracy reflects information already present in the representations, rather than capacity of the probe itself.

---

[2]Details on $N$ are provided in the Appendix.

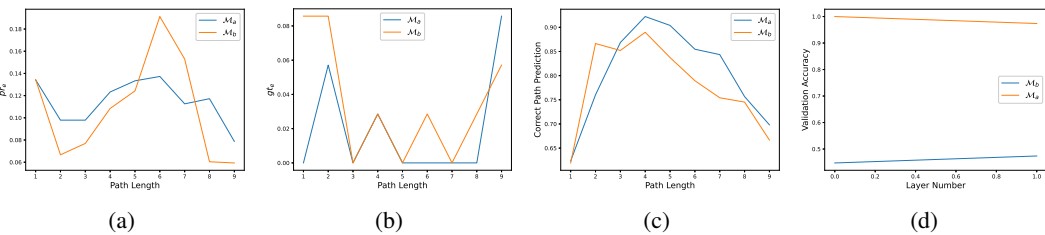

(a)          (b)          (c)          (d)

Figure 1: Measure $pr_e$ and $gt_e$ for the toy graph dataset v pathlength; 1c shows the prediction accuracy of the two models vs the path length in the graph; 1d shows the probe accuracy of the two models vs the layer number in the graph;

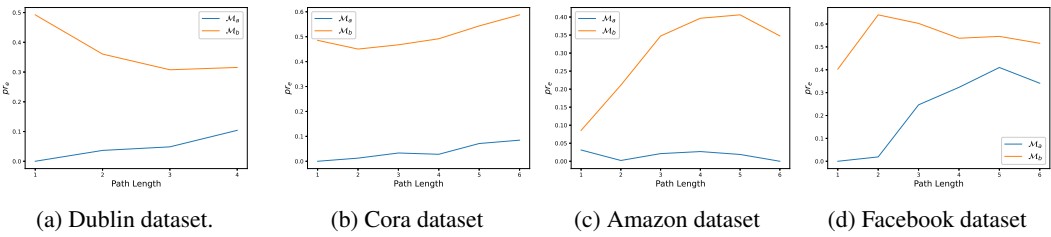

(a) Dublin dataset.          (b) Cora dataset          (c) Amazon dataset          (d) Facebook dataset

Figure 2: Measured $pr_e$ Various datasets.

We construct a balanced dataset of positive pairs $((S, D) \in \{edges\})$ and negative pairs $((S, D) \notin \{edges\})$, sampled from graphs disjoint from the training set. The dataset is split into training and validation sets in an $80 : 20$ ratio.

For each pair, hidden activations are extracted from every attention layer. A separate linear probe is trained for each layer to classify edge presence, providing a layer-wise measure of how neighborhood information is represented.

## 4 PERFORMANCE OF IN-CONTEXT LEARNING VS ZERO SHOT LEARNING

To initially study the performance of in-context Learning and the zero shot learning model we generate 5 small graphs using networkX [3], containing 10 nodes and 20 edges. The training and validation data is generated from each graph dataset[4]. For more details about the training data we refer the reader to Section-3.1.

We evaluate the following models :

1) In-context learning ($\mathcal{M}_a$) : The model receives the relevant graph as context and is queried to predict a path between a pair of nodes. For training, each input sequence also includes the full graph to guide the model's predictions.

2) Zero Shot Learning ($\mathcal{M}_b$) : The model receives only the pair of query nodes and predict a path in the graph without any context. This model is trained specifically on a single graph.

Both $\mathcal{M}_a$ and $\mathcal{M}_b$ are GPT-style transformer models with 2 hidden layers, 8 attention heads, and an embedding size of 128. The models are trained from scratch with randomly initialized weights in an auto-regressive fashion for next-word prediction, ensuring no prior knowledge of the graph structure.

We evaluate the models by computing $pr_e$ and $gt_e$ errors as described previously. These metrics quantify hallucinated edges and missing edges in the reconstructed adjacency matrices, respectively.

---

[3]https://networkx.org/

[4]The training and validation data contains paths from all the graphs.

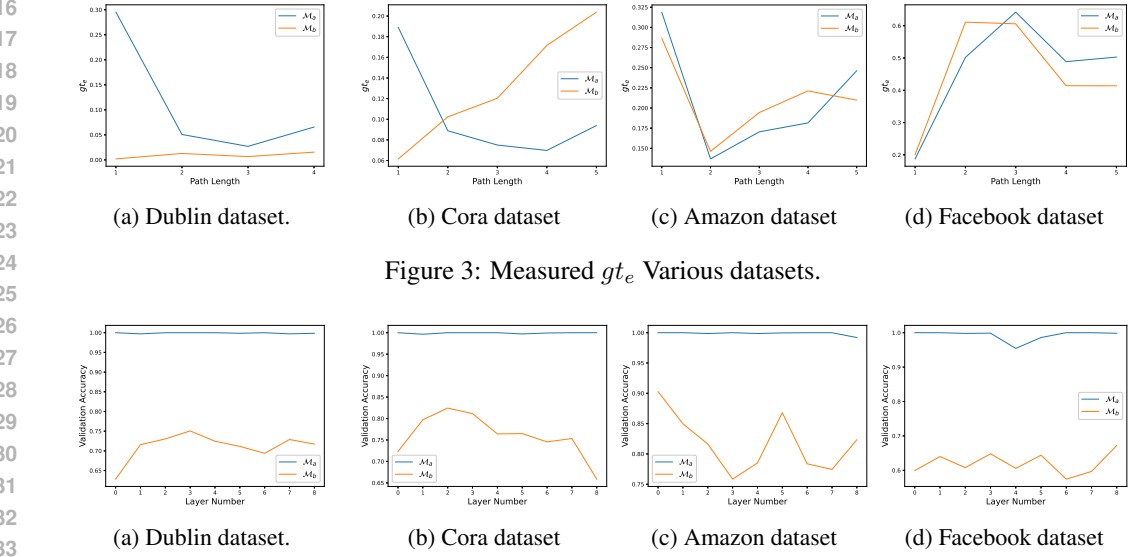

(a) Dublin dataset.     (b) Cora dataset     (c) Amazon dataset     (d) Facebook dataset

Figure 3: Measured $gt_e$ Various datasets.

(a) Dublin dataset.     (b) Cora dataset     (c) Amazon dataset     (d) Facebook dataset

Figure 4: Measured Classification Accuracy of linear probe for Various datasets. x axis represents the layer from which the activations were chosen, the y axis represents the accuracy of the linear probe on validation dataset.

Figures 1a and 1b present the reconstruction errors of both models on the graphs. The results indicate that the two models yield comparable reconstruction errors as well as similar path prediction accuracy.

Figure 1c illustrates the accuracy of predicting the correct path given the query $(S, D, L)$ as the path length increases. We observe that $\mathcal{M}_a$ achieves slightly higher accuracy than $\mathcal{M}_b$ for longer paths, while their overall performance remains largely comparable. This pattern aligns with the reconstruction errors, where both models exhibit a similar level of error.

We also train linear probes for the models for various attention layers. The training and validation data for probes is generated from the graph as described previously. Figure-1d shows the linear probe accuracy v the path length. We observe the probes for model $\mathcal{M}_a$ consistently outperforms the probes for model $\mathcal{M}_b$.

The high performance of the probe on $\mathcal{M}_a$ indicates that the activations of the in-context learning model encode structural information about the graph - specifically, whether two nodes are neighbors - whereas the activations of $\mathcal{M}_b$ do not contain this information.

The results for model $\mathcal{M}_a$ are unsurprising since the relevant graph to the query is provided as input to model, hence it can be reconstructed by a probe at the initial layers. Interestingly the activations of model $\mathcal{M}_b$ fail to support a reliable reconstruction of the graph structure.

## 5 PREDICTION PERFORMANCE: SINGLE VS MULTIPLE MODELS

On the synthetic graph, we observed that the in-context learning model $\mathcal{M}_a$ produced slightly fewer errors and achieved higher path-prediction accuracy than the zero-shot learning model $\mathcal{M}_b$, although the overall performance of the two models was largely comparable. In that setting, the full graph was provided as context to $\mathcal{M}_a$, making the tasks of reconstruction and path prediction relatively straightforward.

In contrast, real-world graphs are substantially larger and more complex, making it impractical to supply the entire graph as input context. Instead, only a small, relevant subgraph is provided. This change in input fundamentally alters the difficulty of the task: while in Section 4 both models performed similarly when $\mathcal{M}_a$ had access to the full graph, in real-world scenarios the length of the subgraph context plays a decisive role. In this section, we evaluate the models on real-world datasets, where only a subgraph relevant to the query is provided as input context. This setup allows

us to examine how limiting the context to local subgraphs, rather than the full graph, affects the in-context learning process.

**Datasets**

We train and test our models on the following datasets :

`CORA Dataset` :   Cora Citation Network is a directed graph where nodes are scientific papers and edges represent citations (i.e., paper A $\rightarrow$ paper B means A cites B). For our experiments we sample 500 nodes from the CORA dataset. We generated 60000 training sequences from the graph.

`Dublin Street Map` :  A real-world directed road network of Dublin, Ireland. We use the Open-street Map API to extract the graph. For our experiments we sample 500 nodes from the original map dataset. We generated 60000 training sequences from the graph.

`Facebook Social Circle Dataset` : A real-world directed graph that dataset consists of 'circles' (or 'friends lists') from Facebook. Each node in a graph denotes an individual and an edge between the individual represent a connection on facebook between the nodes.  For our experiments we sample 500 nodes from the CORA dataset. We generated 60000 training sequences from the graph.

`Amazon Co-Purchase Network` : A graph crawled from amazon website. It is based on Customers Who Bought This Item Also Bought feature of the Amazon website. If a product i is frequently co-purchased with product j, the graph contains an edge from i to j. Each product category provided by Amazon defines each ground-truth community. For our experiments we sample 500 nodes from the CORA dataset. We generated 60000 training sequences from the graph.

Training sequences are constructed as described in Section 3.1, capturing paths between node pairs. In this setting, the context provided to $\mathcal{M}_a$ consists solely of a relevant subgraph extracted around the query nodes.  For more details on how the subgraph is generated we refer the reader to the appendix.

**Experiments**

Models $\mathcal{M}_a$ and $\mathcal{M}_b$ are trained on each dataset using the approach described in the previous section.  Both models have 10 hidden layers, 10 attention heads, and an embedding size of 500 to accommodate the increased graph complexity.

The adjacency matrix is generated from the predictions of sampled $(S,D)$ node pairs as explained previously, that is then used to calculate the errors generated by the models.

Figures 2 and 3 show the errors for both models across datasets for various path-lengths. Model $\mathcal{M}_a$ consistently exhibits lower error rates than $\mathcal{M}_b$. This improvement can be attributed to the explicit subgraph context provided to $\mathcal{M}_a$, which allows it to better capture underlying graph structure and reduce hallucination. These results demonstrate that in-context learning produces fewer error than zero shot learning when applied to real-world graphs.

In the previous setting, where the full graph was available as context both models performed similarly.  However, when only relevant subgraphs are provided, the performance of $\mathcal{M}_a$ improves relative to $\mathcal{M}_b$. This suggests that shorter, localized contexts make in-context learning more effective when holding the model complexity constant, whereas zero-shot learning fails to take advantage of the available structural information.

**Probing Internal State**

We train linear probes for the models for all the attention layers. The training and validation data for probes is generated from the graph as described previously.

Figure 4 shows the validation accuracy of the linear probe for both models as a function of attention layer. Across all layers, the probe trained on model $\mathcal{M}_a$ consistently outperforms the probe trained on model $\mathcal{M}_b$. We expected the earlier layers of $\mathcal{M}_a$ to achieve good probe accuracy, since the input graph information is explicitly provided as context. However, we also observe that this signal propagates through later layers.

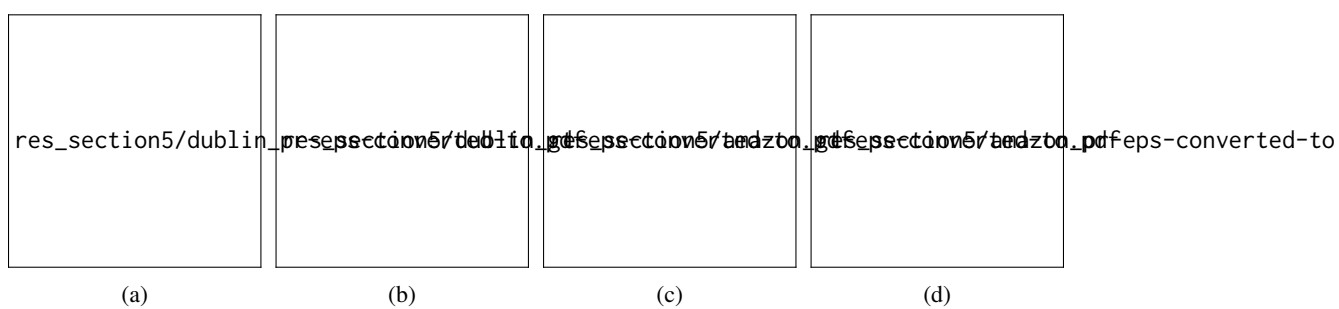

res_section5/dublin_pre.pdfeps-converted-to.pdf res_section5/dublin_gt.pdfeps-converted-to.pdf res_section5/amazon_gt.pdfeps-converted-to.pdf res_section5/amazon_pdfeps-converted-to.p

(a)         (b)         (c)         (d)

Figure 5: Measure $pr_e$ and $gt_e$ on various other datasets for the model trained on cora dataset. 5a and 5b shows the $pr_e$ and $gt_e$ error for dublin dataset; 5c and 5d shows the $pr_e$ and $gt_e$ error for amazon dataset;

## 6 NO EVIDENCE FOR MEMORIZATION BY IN-CONTEXT LEARNING

Our previous experiment shows that model $\mathcal{M}_a$ outperforms model $\mathcal{M}_b$ across various datasets when only a relevant subgraph is provided. The huge performance improvement is due to the extra context provided to the model during training and inference. We also wanted to observe whether $\mathcal{M}_a$ is relying on memorizing its training data or actually learning graph processing patterns that can generalize. To test this, we evaluate the models on paths from a completely different graph: a model trained on sequences from graph $\mathcal{G}_s$ is now tested on a separate graph $\mathcal{G}_d$.

We generate $\mathcal{G}_d$ from a new graph dataset and select a subgraph from $\mathcal{G}_d$ such that the number of nodes are similar to that of $\mathcal{G}_s$. We sort the nodes in both graphs by their total degree and then rename the nodes in $\mathcal{G}_d$ to match the degrees in $\mathcal{G}_s$. This way, the graphs are similar in structure but do not share the same nodes or edges. If $\mathcal{M}_a$ had memorized the training data, we would expect its performance to drop sharply on this new graph.

To reconstruct the adjacency matrix, we again sample $N$ number of $(S, E)$ nodes from the new graph dataset $\mathcal{G}_d$. We then compare it to the true adjacency matrix of $\mathcal{G}_d$ to calculate the prediction errors ($pr_e$ and $gt_e$).

Figure 5 shows the errors for both models when trained on the CORA dataset and tested on the Dublin street map and Amazon co-purchase datasets. We see that $\mathcal{M}_a$ produces slightly fewer errors than $\mathcal{M}_b$ for shorter paths, but as the path length grows, the errors of both models become similar. We observe similar results on other dataset pairs, which are provided in the appendix. Fewer errors for $\mathcal{M}_a$ at short path lengths suggests that it is not simply memorizing the training data, instead it is using the context to generate predictions.

The results suggests that the improved performance of in-context learning comes from using the provided subgraph during prediction, rather than from memorizing training data. However, when tested on completely new graphs, performance decreases, suggesting that the model's ability to generalize depends on patterns seen in the training graphs and does not fully transfer to graphs with different structures.

## 7 SCALING OF IN-CONTEXT LEARNING AND ZERO SHOT LEARNING MODELS

Scaling is a crucial consideration for evaluating whether the observed behaviors of LLMs persist as input context or graph size increases. In this section, we study : (1) the effect of enlarging the subgraph provided as context for in-context learning models ($\mathcal{M}_a$), and (2) the effect of increasing the size of the training graph for zero shot learning models ($\mathcal{M}_b$).

Both models are trained on synthetic graph datasets generated with networkX, where we vary the number of nodes in each graph. For every dataset with a fixed node count, models $\mathcal{M}_a$ and $\mathcal{M}_b$ are trained. The training and validation sets are constructed using the procedure outlined earlier. For

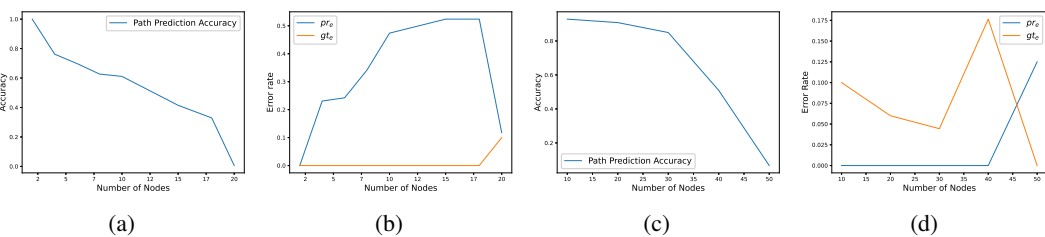

Figure 6: 6a shows the prediction accuracy of the model $\mathcal{M}_a$ as the context is varied; 6b shows the error in prediction of the model $\mathcal{M}_a$ as the context is varied; 6c shows the prediction accuracy of the model $\mathcal{M}_b$ as the graph size is varied; 6d shows the error in prediction of the model $\mathcal{M}_b$ as the graph size is varied;

model $\mathcal{M}_a$, we adjust the number of nodes included in the input context, whereas for model $\mathcal{M}_b$, we adjust the number of nodes in the underlying graph structure.

The models are GPT style models with 2 hidden layers, 8 attention heads and an embedding size of 128 [5]. The training is carried out in an auto-regressive fashion as explained previously.

**Scaling of Context in In-Context Learning** $(\mathcal{M}_a)$**:**

Figure 6a illustrates path-prediction accuracy as the size of the subgraph provided in the context increases, while Figure 6b shows the corresponding reconstruction errors. As the number of nodes in the context grows, path-prediction accuracy decreases and the proportion of hallucinated edges $(pr_e)$ increases. It can be seen that the accuracy of in-context learning model predictions degrades as the context size increases.

**Scaling of Graph Size in Zero shot learning** $(\mathcal{M}_b)$**:**

We also examined the effect of increasing the size of an almost linear underlying training graph. Figures 6c and 6d show that as the graph size increases, prediction accuracy declines and reconstruction error grows. However, compared to $\mathcal{M}_a$, $\mathcal{M}_b$ retains relatively high accuracy even for larger graphs, indicating that zero-shot learning scales more gracefully with graph size.

The two models exhibit distinct behaviors as the number of nodes increases. In-context learning models ($\mathcal{M}_a$) show a decline in path-prediction accuracy and an increase in hallucinated edges as the context size grows, whereas zero-shot learning models ($\mathcal{M}_b$) show decreasing prediction accuracy and increasing reconstruction error as the training graph size increases.

These findings highlight that both approaches have inherent scaling limitations. For in-context learning, performance is sensitive to the size of the input context, while for zero-shot learning, performance is constrained by the size of the training graph.

## 8 CONCLUSION

Our findings indicate that current LLMs do not develop robust internal representations of underlying graph structures from training data alone. In zero-shot settings, these models produce high reconstruction errors and show no evidence of internal structure learning. While providing contextual information improves reconstruction and probing accuracy, this demonstrates that the models primarily rely on the input context rather than forming generalizable, internal world models during training.

We argue that if these models truly learned internal representations from training data, they should demonstrate lower reconstruction and probing errors even in zero shot learning settings. Our findings suggest that current LLMs primarily rely on explicit contextual information rather than developing robust internal world models during training.

---

[5]The trained model had 5.3M parameter

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

# A APPENDIX

## A.1 GENERATING QUERIES

In this section we provide details about how we generated training and test sentences for training the models.

**1.) Training Queries** Training Queries consists of valid and in-valid paths from the graph. A Graph $\mathcal{G}$ consists of nodes and edges, an edge in a graph represents a relationship between two nodes. Sentences from the graph can be generated by randomly selecting an edge from the list of all possible edges, then traversing through that edge until we either reach an "EndNode"[6], or we reach the desired path length, see here for example Vafa et al. (2024). Each full-path traversed can be converted to an input sentence of the following format :

`[StartNode] [EndNode] [Pathlength] [StartNode] [List of Nodes] [END]`.

Where the StartNode is the starting node, and EndNode is the destination node, Pathlength is the number of nodes between the StartNode and the EndNode of the path and [END] token is a special token that is used to indicate end of path.

In addition to the paths that can be reached we sample X% of un-reachable paths from the graph $\mathcal{G}$ as well. The sentences formed in such cases follow the format :

`[StartNode] [EndNode] [Pathlength] [NP] [END]`.

where [NP] is a special token that denotes no path exists between the StartNode and EndNode for the given Pathlength.

**2.) Test Queries** To query the trained LLM we use the following format for our test-queries:

`[StartNode] [EndNode] [Pathlength]`

We generate various test queries for our experiments. Specific details about them can be found in the following sections. The format of the test queries remains the same.

Note. Since for a given Node in a graph $\mathcal{G}$ number of non-neighbour's node always exceeds the number of neighbour nodes we only sample $k$ - (Number of Neighbours for the StartNode), number of non-neighbour's node from the input data.

We also generate a separate set of training and validation queries that contain a subgraph in the context. In such cases the training and validation queries will be updated to the following format :

`State edgeA || edgeB QUERY : [StartNode] [EndNode] [Pathlength] [GEN]`

Where edgeA and edgeB are various edges extracted from the input graph, that is relevant to answer the input queries. [GEN] is a especial token that guides the LLM to only start generated sequences. We refer the reader to Section-**??** for more details about sampling of edges present in the context.

## A.2 RECONSTRUCTING LEARNED GRAPH

Visualizing the world state learned by the model is an open problem. Previously authors have used the prediction capabilities of the LLM to generate the implicit graphs learned by the model Vafa et al. (2024). Another approach is to look at the internal activation of the LLM to generate saliency maps of the world-state learned Karvonen (2024); Li et al. (2024). In this work we will be using the predictions capabilities of the LLM to generate a world-state learned by the LLM.

A graph can be represented as an Adjacency Matrix. It is an NxN matrix, where each row and column denotes a node. Each item at the index $(i, j)$, denotes the presence or absence of an edge between the nodes $i$ and $j$.

The paths predicted from test queries can be used to generate an adjacency matrix. We use Algorithm-1 to generate the adjacency matrix.

---

[6]Node that has no outgoing edge

**Algorithm 1 Generate Adjacency Matrix**; $predictions$ are the generated paths from the model; $N$: Number of nodes in the graph.

**function** $generateMatrix(prediction[\ ],N)$
    set $numPreds = len(predictions)$
    $A[N][N] = 0$                           ▷ Adjacency Matrix N x N having all the values zero's
    **for** $k \leftarrow 1$ to $numPreds$ **do**
        $sent \leftarrow prediction[k]$
        set $M \leftarrow len(sent)$
        **for** $j \leftarrow 1$ to $M - 1$ **do**
            $s_n \leftarrow sent[j]$
            $d_n \leftarrow sent[j + 1]$
            $s_n \leftarrow$ location of source node in adjacency matrix
            $d_n \leftarrow$ location of destination node in adjacency matrix
            **if** $d_n\ != [END] or d_n\ != [NP]$ **then**
                $A[s_n][d_n] = 1$
            **end if**
        **end for**
    **end for**
    **return** $A$
**end function**

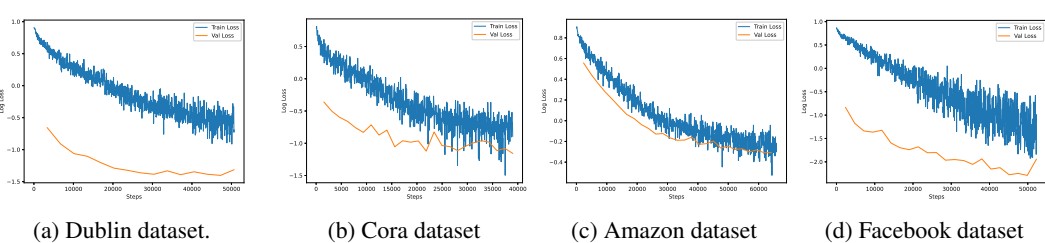

(a) Dublin dataset.          (b) Cora dataset          (c) Amazon dataset          (d) Facebook dataset

Figure 7: Measured train and validation loss for various datasets for model $\mathcal{M}_a$

### A.3 HYPER PARAMETERS FOR GPT TRAINING

All the models were trained in an auto regressive fashion using pytorch and pytorch lightning with the following hyper parameters - batch_size = 4, learning rate = 0.0001. Adam optimizer was used to train the model.

### A.4 TRAINING THE GPT MODEL

Figure-7 and Figure-8 shows the training and validation loss for models $\mathcal{M}_a$ and $\mathcal{M}_b$ respectively.

### A.5 TRAINING THE LINEAR PROBE

We train a logistic-regression model provided by scikit-learn[7] on the attention layer of the trained model.

### A.6 PERFORMANCE OF $\mathcal{M}_a$ ON VARIOUS DATASETS

Figure 11, Figure 10 and Figure 9 shows the reconstruction errors of the model that are validated on new graphs. We observe that model $\mathcal{M}_a$ produced fewer errors than model $\mathcal{M}_b$.

### A.7 EXTRACTING SUBGRAPH FOR $\mathcal{M}_a$

We construct a subgraph from the original graph that is relevant to the input query. This subgraph includes the query-specific path as well as a subset of additional edges.

---

[7]https://scikit-learn.org/stable/modules/generated/sklearn.linear_model.LogisticRegression.html

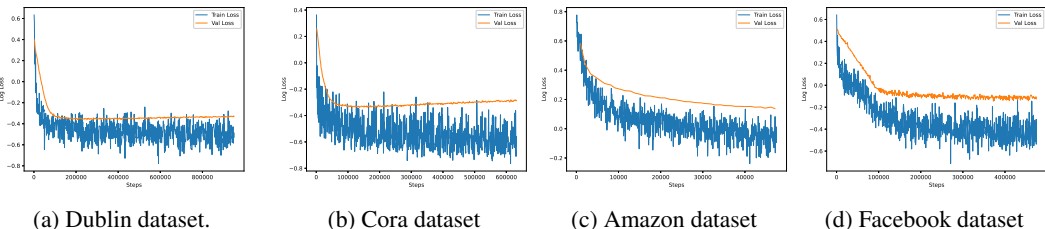

(a) Dublin dataset.  (b) Cora dataset  (c) Amazon dataset  (d) Facebook dataset

Figure 8: Measured train and validation loss for various datasets for model $\mathcal{M}_b$

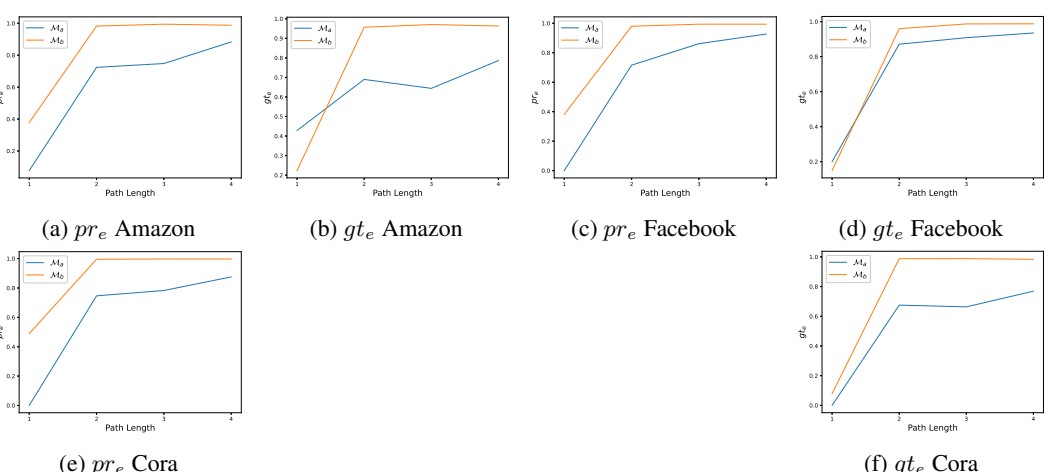

(a) $pr_e$ Amazon  (b) $gt_e$ Amazon  (c) $pr_e$ Facebook  (d) $gt_e$ Facebook

(e) $pr_e$ Cora  (f) $gt_e$ Cora

Figure 9: Measure $pr_e$ and $gt_e$ on various other datasets for the model trained on dublin dataset.

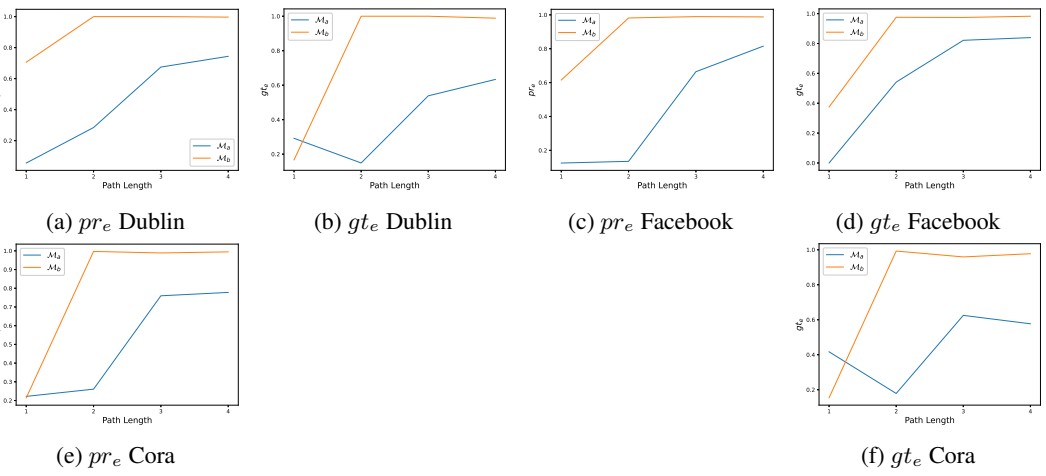

(a) $pr_e$ Dublin  (b) $gt_e$ Dublin  (c) $pr_e$ Facebook  (d) $gt_e$ Facebook

(e) $pr_e$ Cora  (f) $gt_e$ Cora

Figure 10: Measure $pr_e$ and $gt_e$ on various other datasets for the model trained on amazon dataset.

To generate the subgraph, we first identify all neighbors of the nodes that are relevant to the input query. From the resulting candidate subgraph, we then randomly discard 60% of the edges that are not directly related to the query. The resulting pruned subgraph, which preserves both the essential path and a controlled number of auxiliary edges, is subsequently incorporated into the input.

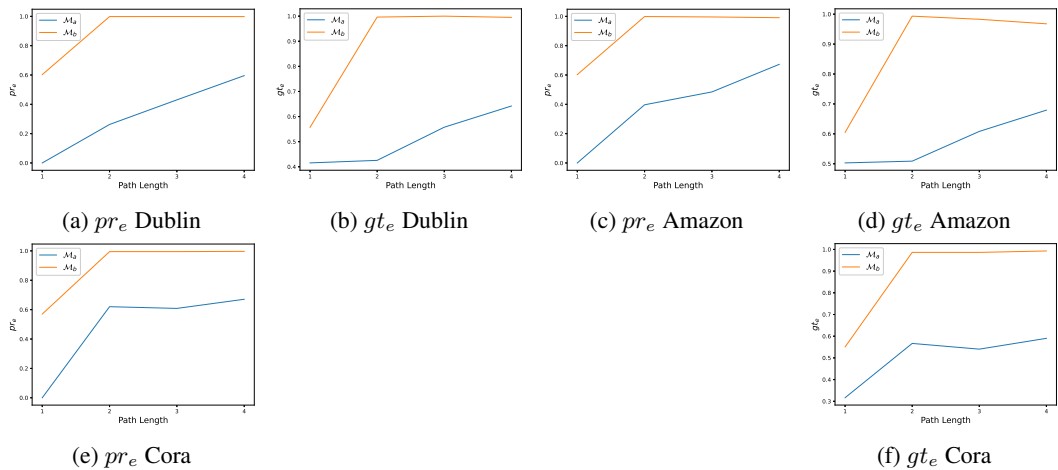

(a) $pr_e$ Dublin     (b) $gt_e$ Dublin     (c) $pr_e$ Amazon     (d) $gt_e$ Amazon

(e) $pr_e$ Cora     (f) $gt_e$ Cora

Figure 11: Measure $pr_e$ and $gt_e$ on various other datasets for the model trained on Facebook dataset.

## A.8 THE USE OF LARGE LANGUAGE MODELS (LLMS)

A Large Language Model, specifically GPT-5, was used to assist in writing the manuscript and refining the English. It was not involved in other stages of the project, including ideation, experimentation, or data interpretation.