# OpenReview forum: "Do LLMs Learn Graph Representations Without Context?"
_ICLR.cc/2026/Conference — ICLR 2026 Conference Withdrawn Submission_

### Official Review · Reviewer_cmzv · 2025-10-28

**Soundness:** 2
**Presentation:** 1
**Contribution:** 1
**Rating:** 2
**Confidence:** 4

**Summary:**

In this paper, the author analyzes how well the LLMs can internalize structural knowledge like graphs, under both in-context learning and zero-shot learning settings. Authors discover that in-context learning is crucial for the model to accurately recover the learned graph structure.

**Strengths:**

The writing is straightforward and easy to follow.

**Weaknesses:**

- The manuscript is far from complete. It contains unrendered pictures (Figure 5), a missing reference (line 578), etc.
- The author claims to analyze the behavior of LLMs. However, all experiments are conducted by training a tiny transformer model from scratch. Therefore, it is unknown whether a similar conclusion can be made for real LLMs.
- To me, it is straightforward that in-context learning can help to recover the path/edge as you provided such context to the model. I do not quite understand what the authors are trying to claim. The experiments seem to just show that transformer models can utilize the context information, which is quite common sense for the community.

**Questions:**

See above.

---

### Official Review · Reviewer_ftvK · 2025-10-28

**Soundness:** 1
**Presentation:** 1
**Contribution:** 2
**Rating:** 2
**Confidence:** 5

**Summary:**

The paper propose to use probing on GPT-like model trained for graph edge prediction to test if the model has internal ability to learn from the graph. Through probing on various graph datasets, they show that the model has minimal ability to internalize graph knowledge.

**Strengths:**

The idea is interesting, using techinques for language to examine the capability of graph models.

**Weaknesses:**

- The paper is unfinished with missing figures in page 8.

- The system is tested only on a very small model, the internalization can be highly correlated with data and model size, this is be considered.

- Testing only on graph structure can be limited, as some internalization can happen as a synergitic process between feature on the graph and the structure.

- The paper possess interesting idea, but there are too many unanswere question and it is not ready for publication.

**Questions:**

N/A

---

### Official Review · Reviewer_gZfy · 2025-10-29

**Soundness:** 2
**Presentation:** 1
**Contribution:** 2
**Rating:** 2
**Confidence:** 4

**Summary:**

The paper asks whether GPT-style sequence models can acquire **internal graph representations** without explicit contextual input. The authors train two small transformer models from scratch on path sequences sampled from graphs: **Ma (in-context)**, which receives a query plus a relevant subgraph as input, and **Mb (zero-shot)**, which receives only the query node pair and path length. They evaluate via **adjacency reconstruction**—reporting hallucinated edges (**pre**) and missing edges (**gte**)—and **linear probes** on hidden states to test if neighborhood information is encoded. On small synthetic graphs where Ma is given the **full graph** as context, Ma and Mb exhibit **comparable** reconstruction errors and path accuracy, with Ma slightly better for longer paths. On real-world datasets (CORA, Dublin, Facebook, Amazon) where Ma gets only a **local subgraph**, Ma shows **lower pre/gte** and higher probe accuracy across layers than Mb. Cross-graph tests suggest Ma’s gains are not just memorization, and scaling studies show Ma’s accuracy degrades as context grows, while Mb degrades with graph size but somewhat more gracefully. The authors conclude that, in these settings, models **rely on explicit context** for structural reasoning and do **not** develop robust internal graph structure purely from training data.

**Strengths:**

- **Clear experimental framing and metrics.** The reconstruction metrics (pre/gte) and layer-wise linear probing provide complementary views of what is in the output vs. internal activations.
- **Nuanced results across regimes.** The paper carefully distinguishes the synthetic/full-context case (Ma ≈ Mb) from real-world/subgraph context (Ma > Mb), avoiding overgeneralization.
- **Generalization and scaling analyses.** Cross-graph evaluation argues against simple memorization, and the context-size vs. graph-size scaling experiments clarify where each regime breaks.

**Weaknesses:**

- **Presentation and clarity.** The overall presentation needs substantial improvement: key terms (e.g., “zero-shot” vs. “in-context”) are introduced late or used inconsistently; the data pipeline and subgraph construction are hard to follow without a schematic; several figures/tables lack self-contained captions and clear definitions of metrics (e.g., `pre`/`gte`); and experimental details (model size, training budget, seeds) are scattered. Tightening organization, defining terminology up front, adding a one-figure overview of the training/evaluation pipeline, and centralizing experimental settings would make the contributions much clearer.
- **Terminology and scope of “zero-shot.”** Both Ma and Mb are **trained from scratch** on graph-generated text; “zero-shot” is used to denote **no contextual examples at inference**, not pre-trained LLM zero-shot usage. A brief clarification would prevent confusion with standard NLP usage.
- **Proxy for “internal representation.”** Adjacency reconstruction from predicted paths (with a reduced reference matrix) and linear probes are sensible, but they remain proxies; explicitly discussing limits of these proxies would strengthen the claims.
- **Context construction details matter.** The subgraph for Ma is created by including neighbors and **randomly discarding ~60% of non-query edges**; this design choice could affect difficulty and should be motivated/ablationed.
- **Model/data scale.** Results are based on relatively small transformers (e.g., ~5.3M parameters) and sampled 500-node subgraphs/datasets; the conclusion about “LLMs” might be framed as **“in our small GPT-style models and data regime”**.

**Questions:**

Please refer to the weaknesses.

---

### Note · Authors · 2025-11-14

I have read and agree with the venue's withdrawal policy on behalf of myself and my co-authors.